# Deep Reinforcement Learning for Efficient and Fair Allocation of Healthcare Resources

## Abstract

Scarcity of health care resources could result in the unavoidable consequence of rationing. For example, ventilators are often limited in supply, especially during public health emergencies or in resource-constrained health care settings, such as amid the pandemic of COVID-19. Currently, there is no universally accepted standard for health care resource allocation protocols, resulting in different governments prioritizing patients based on various criteria and heuristic-based protocols. In this study, we investigate the use of reinforcement learning for critical care resource allocation policy optimization to fairly and effectively ration resources. We propose a transformer-based deep Q-network to integrate the disease progression of individual patients and the interaction effects among patients during the critical care resource allocation. We aim to improve both fairness of allocation and overall patient outcomes. Our experiments demonstrate that our method significantly reduces excess deaths and achieves a more equitable distribution under different levels of ventilator shortage, when compared to existing severity-based and comorbidity-based methods in use by different governments. Our source code is included in the supplement and will be released on Github upon publication.

## 1 Introduction

The Institute of Medicine (IOM) defines Crisis Standards of Care as "a substantial change in usual health care operations and the level of care it is possible to deliver, which is made necessary by a pervasive (e.g., pandemic influenza) or catastrophic (e.g., earthquake, hurricane) disaster" (Gostin et al., 2012). These guidelines recognize that pandemics such as COVID-19 can strain health systems into an absolute scarcity of health care resources and could result in the unavoidable consequence of rationing.

Following the IOM framework, state governments throughout the U.S. have developed allocation protocols for critical care resources during the pandemic (Piscitello et al., 2020). Consistent with the broad consensus of ethicists and stakeholders (Emanuel et al., 2020), these protocols aim to triage patients via a pre-specified, transparent, and objective policy. Despite this common general framework, critical details of these protocols vary widely across the U.S. (Piscitello et al., 2020). Many protocols aim to maximize total benefits to the population by ranking patients with objective outcome predictors like the Sequential Organ Failure Assessment (SOFA) score (Vincent et al., 1996), the conversion of SOFA into "priority scores" differs substantially between protocols. The protocols also vary on whether they prioritize younger patients or those without pre-existing medical conditions. For example, the SOFA protocol (e.g., used in New York (VEN, 2015)) is focused on saving the most lives in the short term and ignores both age and pre-existing medical conditions. In contrast, the multiprinciple protocols (e.g., used in Maryland and Pennsylvania) also prioritize younger patients and those without pre-existing medical conditions (Biddison et al., 2019). The fact that these protocols are heuristic based and substantial variation among the protocols calls for evidence-based protocol design using machine learning. The allocation protocols are typically assessed on a daily basis, making it a sequential decision making process and suitable for reinforcement learning (RL).

In this study, we investigate the use of RL for critical care resource allocation policy optimization. Our goal is to learn an optimal allocation policy based on available observations, in order to maximize lives saved while keeping rationing equitable across different races. It is quite natural that sequential allocation decisions on critical care resources can be modeled by a Markov decision process (MDP),

and the allocation protocols by RL policies. We use a large, diverse, multi-hospital population of critically ill patients amid COVID-19 pandemic to evaluate our proposed method and comparison models.

However, application of reinforcement learning (RL) to this problem is nontrivial for practical considerations. In particular, empirical assessment of their performance suggested that most such protocols and their variants used earlier in US states were significantly less likely to allocate ventilators to Black patients (Bhavani et al., 2021). In addition, the exact SOFA score is shown to be only modestly accurate in predicting mortality in COVID-19 patients requiring mechanical ventilation (Raschke et al., 2021). There is a vital need to improve both the utility and equity of these allocation protocols to fairly and effectively ration resources to critically ill patients. Our main contributions are summarized as follows:

1. We are the first to formulate fair health care resource allocation as a multi-objective deep reinforcement learning problem. The goal is to integrate the utilitarian and egalitarian objectives in the RL rewards.

2. We propose a Transformer-based deep Q-network capable of making allocation decisions by incorporating the disease progression of individual patients and the interaction effects among patients during the critical care resource allocation.

3. We apply our approach to real-world clinical datasets. Experiments show that our approach leads to fair allocation of critical care resources among different races, while maintaining the overall utility with respect to patient survival.

## 2 RELATED WORK

Health care resources, including personnel, facilities, medical equipment, etc., are never in excess, especially amid pandemics (Emanuel & Wertheimer, 2006) or in the case of intensive care (Truog et al., 2006). Decision-making is vital for health care outcome optimization (Persad et al., 2009), bearing the scarcity of resources. A successful strategy should account for optimal resource utilization, equity and fairness, cost-effectiveness, and crisis buffering. Health care is a fundamental human right, and any resource allocation strategy should not disproportionately disadvantage or discriminate against specific groups. However, implicit or explicit discrimination has permeated health care and medicine with a long history, presenting numerous instances of biased outcomes (Dresser, 1992; Tamayo-Sarver et al., 2003; Chen et al., 2008).

Due to its inherent nature of comprehending goal-oriented learning and decision-making challenges, RL possesses the potential to generate optimized allocation strategies. However, Machine learning approaches, including RL, can be biased towards favoring the accuracies of majority classes at the expense of minority classes, and do not automatically align with fairness objectives Mehrabi et al. (2021). In fact, RL may further exacerbate disparities across patient groups by ignorance of fairness considerations Liu et al. (2018). In recent years, there have been growing literature for fairness in machine learning across various fields (Mehrabi et al., 2021; Pessach & Shmueli, 2022; Binns, 2018 ), including health care (Ahmad et al., 2020; Rajkomar et al., 2018; Pfohl et al., 2021; Wang et al., 2022; Li et al., 2022). However, there lack considerations of fairness of RL algorithms in the application of health care resource allocation.

Previous studies employed RL to simulate the trajectory of a pandemic to facilitate the adoption of different levels of lockdown strategies (Zong & Luo, 2022), to infer the future state-level demand for ventilators to inform resource allocation (Bednarski et al., 2021), to efficiently allocate limited PCR tests to screen incoming travelers for possible COVID-19 infections (Bastani et al., 2021) etc. See survey Yu et al. (2021) for more comprehensive summary of RL application in health care. Previous studies centered around the application of RL in the simulation of future trajectories and focus on utilitarian goals such as maximizing efficiency of allocation, which lacked insight into RL's impact on the equity among patients. Given empirical evidences on resulted disparities from in-use critical care resource allocation protocols (Bhavani et al., 2021), the egalitarian aspects of goals deserve focused attentions. However, using RL for simultaneously reducing disparities across different races during critical resource allocation and improving overall patients' outcomes has been underexplored.

## 3 PROBLEM FORMULATION

We formulate the ventilator allocation problem as a day-to-day sequential decision problem. In the event of a ventilator shortage, some patients who require a ventilator may not receive one due to limited availability. In such cases, a triage protocol must be applied to prioritize the dispatching of ventilators to those patients who truly need them, as determined by the doctors. Following existing clinical literature (e.g., Bhavani et al. (2021)), we assume that patients who needed ventilators but did not receive one will die. Thus, there will be no patients *waiting* for a ventilator, as the inability to receive ventilation will result in their immediate removal from the dataset. Also following Bhavani et al. (2021), we do not consider in the main results the withdrawal of a ventilator from one patient to save another, which is a practice drawing extensive debates without widely agreed consensus on principles (e.g., ranging from no mechanism in the Maryland protocol to an explicit SOFA based approach in the New York protocol). But we did include this setting in appendix A.3, in order to provide a comprehensive picture across the spectrum connecting both extremes. In Appendix A.4, we also explored scenarios where patients not being allocated ventilators does not lead to immediate fatalities, which validates the broader applicability of our proposed formulation and model.

The medical condition of a patient at one time point is represented by a $k$-dimensional vector $x \in [0,1]^k \subseteq \mathbb{R}^k$, including demographics, current SOFA score constructs, vital signs, COVID-19 status etc. Assume the intensive care units have up to $N$ patients arriving in a single day, with each being assigned a bed. Here we only consider ventilator scarcity, but bed scarcity can be analogously modeled. We provide a detailed illustration of our formulation and proposed model in Figure. 1. In the following, we give a rigorous formulation of the RL model for both cases of without and with the consideration of the fairness in distributing ventilators.

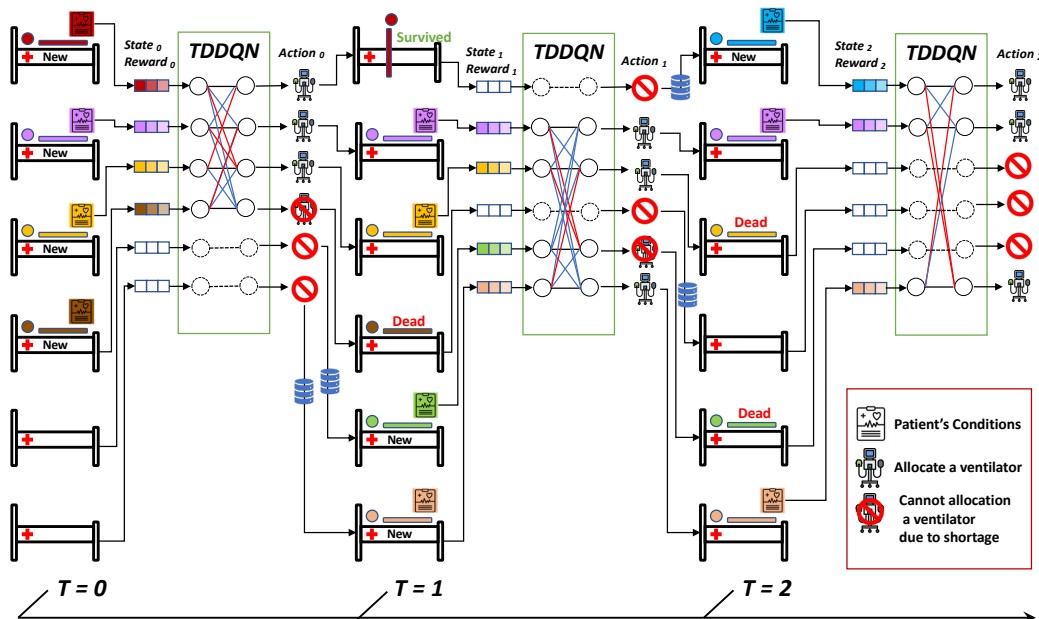

Figure 1: An illustration of the trajectory in our study formulation. Each color represents a separate patient.

### 3.1 STATE SPACE

In the case without consideration of fairness, the state space $\mathcal{S}$ describes the current clinical conditions of all patients in the hospital, as well as their ventilation status. Each state is represented by

$$s = [x_1, x_2, \cdots, x_N, I_1, I_2, \cdots, I_N] \in \mathcal{S} \subseteq \mathbb{R}^{kN+N}, \tag{1}$$

where $x_i \in \mathbb{R}^k$ denotes the current medical condition of the patient on bed $i$, and indicator $I_i \in \{0,1\}$ denotes whether the bed $i$ has been ventilated. Apart from normal medical conditions of patients,

we consider three special conditions: the "SURVIVED", "DEAD", and "VACANT" conditions, corresponding to the cases where the patient in this bed is recovered or dead after ventilation, as well as currently no patient in this bed. They act as terminators for patients or separators between patients in the same bed. Such designs separate different patients on the same bed explicitly, so that the tasks of learning the progression of medical conditions given ventilation and the task of recognizing the end of each patient can be decoupled.

In the case with consideration of fairness, we further record the cumulative numbers of total and ventilated patients of different ethnoracial groups in the state vector, denoted by $n_k, m_k \in \mathbb{R}$ respectively, where $k \in \{\text{'B', 'W', 'A', 'H'}\}$ denotes 4 ethnoracial group in the dataset: non-Hispanic Black, non-Hispanic White, non-Hispanic Asian, Hispanic. Thus, each state

$$s = [x_1, x_2, \cdots, x_N, I_1, I_2, \cdots, I_N, n_B, n_W, n_A, n_H, m_B, m_W, m_A, m_H] \in \mathcal{S} \subseteq \mathbb{R}^{kN+N+8}$$

describes the medical and ventilation status of all current patients, as well as the number of cumulative total and ventilated patients of different ethnic groups.

## 3.2 ACTION SPACE

For both cases, the action space $\mathcal{A}$ is a discrete space denoting whether each bed is on ventilation or not. Let us use 1 to denote ventilate and 0 otherwise, so the action space $\mathcal{A} \subseteq \{0,1\}^N$. Note that we have two constraints on actions:

1. Maximum number of ventilators: $\mathcal{A} \subseteq \{a \in \{0,1\}^N : \sum_{i=1}^{N} a_i \leq B\}$, where $B$ is the number of ventilators the hospital has.

2. Continuous ventilation: a patient who was ventilated must keep ventilation until recovery or death: $\mathcal{A} \subseteq \{a \in \{0,1\}^N : a_i = 1 \text{ if } I_i = 1, i = 1, 2, \cdots, N\}$, where $I_i$ in the state information denotes whether current patient on bed $i$ has been ventilated before.

Therefore, the action space is shrunk to

$$\mathcal{A} = \{a \in \{0,1\}^N : \sum_{i=1}^{N} a_i \leq B \text{ and } a_i = 1 \text{ if } I_i = 1, \forall i = 1, 2, \cdots, N\}. \quad (2)$$

## 3.3 TRANSITION MODEL

For ease of notation, we denote the three special conditions of "SURVIVED", "DEAD" and "VACANT" as $\mathbf{1}, -\mathbf{1}, \mathbf{0} \in \mathbb{R}^k$ respectively. In the case without fairness consideration, given current state $s = [x_1, x_2, \cdots, x_N, I_1, I_2, \cdots, I_N]$ and action $a = [a_1, a_2, \cdots, a_N]$, it will transit to the next state $s' = [x'_1, x'_2, \cdots, x'_N, I'_1, I'_2, \cdots, I'_N]$ in the following coordinate-wise way:

1. If $x_i \neq \mathbf{0}, \mathbf{1}, -\mathbf{1}$, then the patient will transit to "DEAD" condition if not ventilated:

$$P_i^{\text{clinic}}(x'_i|s, a) = \begin{cases} 1 & \text{if } a_i = 0, x'_i = -\mathbf{1}, \\ 0 & \text{if } a_i = 0, x'_i \neq -\mathbf{1}, \\ p^{\text{on}}(x'_i|x_i) & \text{if } a_i = 1, x'_i \neq \mathbf{0}, \\ 0 & \text{if } a_i = 1, x'_i = \mathbf{0}, \end{cases} \quad (3)$$

where $p^{\text{on}}(x'_i|x_i)$ denotes the probability of a patient transiting from medical condition $x_i$ to $x'_i$ given ventilation. The ventilation status is naively transited $P_i^{\text{vent}}(I'_i|s, a) = 1$ if $I'_i = a_i$ and 0 otherwise.

2. If $x_i = \mathbf{0}, \mathbf{1}$ or $-\mathbf{1}$, the action does not influence the transition dynamic:

$$P_i^{\text{clinic}}(x'_i|s, a) = \begin{cases} 0 & \text{if } x'_i = \mathbf{1} \text{ or } -\mathbf{1}, \\ 1 - q_i(s) & \text{if } x'_i = \mathbf{0}, \\ q_i(s) \cdot \xi(x'_i) & \text{if } x'_i \neq \mathbf{0}, \mathbf{1}, -\mathbf{1}, \end{cases} \quad (4)$$

where $\xi(\cdot)$ denotes the distribution of the initial medical condition of a patient when admitted to the critical care units, and $q_i(s)$ denotes the probability of a new incoming patient staying

in bed $i$. Note $q_i(s)$ depends on how new patients are distributed to empty beds. It satisfies $q_i(s) = 0$ if the bed $i$ is already occupied and $\sum_{i=1}^{N} q_i(s) = E \sim Pois(\Lambda)$ assuming the number of incoming patients $E$ obeys an Poisson distribution with parameter $\Lambda$. The ventilation status does not matter at this point, we can set $P_i^{\text{vent}}(I_i'|s,a) = 1$ if $I_i' = 0$ and 0 otherwise.

Given above discussion for transition dynamics of each individual patients/bed, the overall transition probability can be written as

$$P = (P_0^{\text{clinic}}, \cdots, P_N^{\text{clinic}}, P_0^{\text{vent}}, \cdots, P_N^{\text{vent}}). \tag{5}$$

In the case with fairness consideration, we further consider the progress of the number of cumulative patients and ventilated patients of each ethnic group, whose deterministic transitions are naive by their definitions.

## 3.4 REWARD

The reward function consists of the following three parts:

1. Ventilation cost $\mathbf{R_v}$: if a bed uses a ventilator, it occurs a small negative reward $c_1$:

$$\mathbf{R_v}(s,a) = c_1 \cdot \sum_{i=1}^{N} a_i. \tag{6}$$

2. Terminal condition $\mathbf{R_t}$: If a patient is discharged alive after being on a ventilator, a positive reward of 1 is given. If a patient requires a ventilator but is not able to receive one, or dies after being on a ventilator, a penalty of -1 is given:

$$\mathbf{R_t}(s,a) = \sum_{i=1}^{N} \mathbf{1}[x_i = \mathbf{1}] - \sum_{i=1}^{N} \mathbf{1}[x_i = -\mathbf{1}]. \tag{7}$$

3. Fairness penalty $\mathbf{R_f}$: in the case with fairness consideration, we consider the cumulative total and ventilated patients of different ethnic groups, we hope the distribution of ventilators is fair in terms of the proportion of ventilated patients of all groups. Therefore, a penalty of KL-divergence between the frequency distributions of incoming patients of different ethnoracial groups $\mathcal{D}_n \sim [n_B, n_W, n_A, n_H]/(n_B + n_W + n_A + n_H)$ and ventilated patients of different ethnoracial groups $\mathcal{D}_m \sim [m_B, m_W, m_A, m_H]/(m_B + m_W + m_A + m_H)$ is considered:

$$\mathbf{R_f}(s,a) = \text{KL}\left(\mathcal{D}_n \| \mathcal{D}_m\right). \tag{8}$$

Thus, the reward function is given by

$$R(s,a) = \mathbf{R_v}(s,a) + \mathbf{R_t}(s,a) + \lambda \cdot \mathbf{R_f}(s,a), \tag{9}$$

where the parameter $\lambda \geq 0$ balances the trade-off between ventilation effectiveness and fairness. In the case without fairness consideration, we set $\lambda = 0$.

## 4 METHOD

In our problem formulation, the transition model is decomposable: the distribution of the next state of the $i$-th patient $s_i'$ only depends on the patient's individual current state-action $(s_i, a_i)$ and is independent of other patients. This structure simplifies the problem to some extent, as the transitions of the patients' states can be modeled and learned separately. Nevertheless, due to the resource constraints imposed through the restricted action sets and the fairness requirements via reward penalties, the interaction effects among the patients need to be considered. However, modeling all the patients jointly creates computational challenges, as the action space is combinatorial. Also, naively concatenating all patients' state vectors introduces an order among patients, which may cause the model to learn artificial factors on the order of the patients and potentially biases the model.

To circumvent the computational intractability without losing the consideration of the interaction effects among patients, we propose Transformer Q-network parametrization, which inherits the classical Q-learning framework with Q-network parametrization and greedy action selection tailored to our problem structure. Specifically, the Q-network is parametrized as $Q_\theta : \mathbb{R}^{\dim(\mathcal{S})} \to \mathbb{R}^{N \times 2}$, whose input is the state vectors. The $i$-th row of the output $Q_\theta(s)_i \in \mathbb{R}^2$ corresponds to the Q-value contributed by the $i$-th patient given the current state $s$ and the action on the $i$-th patient. We adopt an additive form of the joint Q-value, which estimates the trade-off between effectiveness and fairness when allocating under the constraints and considering the clinical conditions of all patients::

$$Q(s, a) = \sum_{i=1}^{N} Q_\theta(s)_{i,a_i}.$$

In this parametrization, the greedy action $a^* = \arg\max_{a \in \mathcal{A}} Q(s, a)$ can be solved efficiently via finding the extra $(B - |I|)$-largest indices of $\{(Q_\theta(s)_{i,1} - Q_\theta(s)_{i,0}) | 1 \le i \le N, I_i = 0\}$:

$$a_i^* = 1 \Leftrightarrow I_i = 1 \qquad (|I| \text{ patients already in ventilation}) \qquad (10)$$

$$\text{or } s : I_i = 0 \text{ and } Q_\theta(s)_{i,1} - Q_\theta(s)_{i,0} \ge 0 \text{ and } Q_\theta(s)_{i,1} - Q_\theta(s)_{i,0} \text{ ranks top-}(B - |I|) \qquad (11)$$

$$(\text{new } (B - |I|) \text{ patients most in need of ventilation})$$

where the action set $\mathcal{A}$ are defined in Eq. 2.

In our implementation, we use Transformer (Vaswani et al., 2017) for $Q_\theta$. Transformers without positional encodings are permutationally-invariant and can handle inputs with arbitrary sizes. As our input to the model is essentially an orderless set of patients with indefinite size, Transformers are a natural fit for our problem. Each Transformer layer is composed of an MLP layer and an attention layer. The MLP layer acts independently on each element of the input as a powerful feature extractor, while the attention layer is able to capture the interaction effects between all the elements. We combine the **T**ransformer q-network parametrization with **D**ouble-**DQN** (Van Hasselt et al., 2016), a modification to the original DQN (Mnih et al., 2013) capable of reducing overestimation. We refer to the model without and with fairness rewards as **TDDQN** and **TDDQN-fair**, respectively.

## 5 EXPERIMENTS

### 5.1 DATASET

The dataset is sourced from a comprehensive and integrated repository of all clinical and research data sources, including electronic health records, pathology data from multiple real-world hospitals and research laboratories. We collected all the ICU admissions that have been allocated mechanical ventilator between March 15, 2020 and January 15, 2023. We filtered the patients with age between 18 and 95 for consideration. We removed admissions with a ventilator allocation duration exceeding 30 days to eliminate anomalies in the data. We extracted 38 features for each admission, including SOFA components, vital signs, demographics, commorbidities (see appendix A.1 for more details). We will share the de-identified dataset upon publication of the paper.

We split our data into 3 splits, admissions between March 15, 2020 and July 14, 2021 were used as training data; admissions between July 15, 2021 and October 14 2021 were used as validation data on which we selected the best hyper-parameters for testing; admissions between October 15, 2021 and January 15, 2023 were used as test data. The patient distribution of different races are summarized in Table 1. The ventilator demands on each day are shown in Figure 2. We assume that the original dataset obtained from health systems reflects a scenario where there was an abundant supply of ventilators available. We aim to investigate the effectiveness of various protocols in mitigating excess deaths in the event of a reduced number of available ventilators.

### 5.2 TECHNICAL DETAILS

To train the TDDQN model, we developed a simulator by sampling patients' trajectories from the training set. This simulator maintained the clinical trajectory of each patient while randomizing their relative admission order. At each time step, we sampled the initial condition of $E$ patients from the

Table 1: Summary statistics of our dataset. For rows 'Train', 'Validation', 'Test' and 'Overall', N(P%) are the number of patients and the percentage out of all races. For rows 'Female' and 'Deaths', N(P%) are the number of patients and the percentage of females and deaths within that race. $\pm$ denotes std. 'Vent days' is the number of days a patient has been allocated a ventilator.

|  | Asian | Black | Hispanic | White | All races |
|---|---|---|---|---|---|
| Train | 206 (3.8%) | 871 (16.0%) | 668 (12.2%) | 3381 (62.0%) | 5455 |
| Validation | 34 (3.2%) | 161 (15.4%) | 109 (10.4%) | 663 (63.3%) | 1047 |
| Test | 239 (4.5%) | 704 (13.4%) | 497 (9.4%) | 3465 (65.7%) | 5271 |
| Overall | 479 (4.1%) | 1736 (14.7%) | 1274 (10.8%) | 7509 (63.8%) | 11773 |
| Age | 62.3±16.0 | 57.5±16.2 | 55.9±15.8 | 64.2±14.7 | 62.0±15.5 |
| Female | 184 (38.4%) | 814 (46.9%) | 495 (38.9%) | 2848 (37.9%) | 4630 (39.3%) |
| Deaths | 122 (25.5%) | 409 (23.6%) | 358 (28.1%) | 1870 (24.9%) | 2980 (25.3%) |
| Vent days | 4.9±6.1 | 5.9±6.7 | 6.5±7.3 | 4.3±5.3 | 4.8±5.9 |

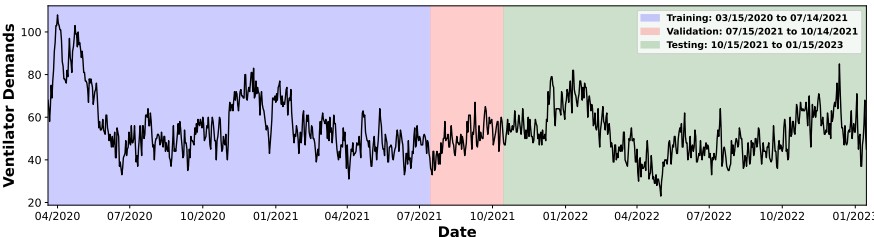

Figure 2: Daily Ventilator Demands

training cohort without replacement. The value of $E$ is determined by a Poisson distribution with $\Lambda = 12$, as inferred from the distribution of ventilator requests in the training cohort. Patients who are not allocated with ventilators or those who are discharged (either alive or deceased) were removed from the ongoing simulator and returned to the sampling pool. Patients who were allocated ventilators progressed to the conditions at the next time step of their own trajectory. This simulator allowed us to simulate and train the model by incorporating realistic patient trajectories and iteratively updating our models in the offline setting. We derived 85 capacity-specific models, considering the maximum daily demand in the validation and testing sets to be 85. The ventilator capacity is normalized from the range of [0, 85] to the range of [0%, 100%].

We compared the performance of the trained protocol with other protocols in allocating ventilators using real-world trajectories from our evaluation and testing dataset. We evaluated the performance of our proposed protocol based on normalized survival rates, with the survival number at no shortage in ventilators as 100% and no patients alive when no ventilator is available as 0%. Fairness was quantified by comparing the allocation rate across four ethnoracial groups. The allocation rate was calculated by dividing the total number of ventilators allocated by the sum of the ventilators requested. The demographic parity ratio (DPR) (Bird et al., 2020) served as the group metric of fairness, and it is defined as the ratio between the smallest and largest group-level allocation rate. A DPR value close to 1 signifies an equitable allocation, indicating a non-discriminatory distribution among the different groups. Additionally, to gain insights into the impact of protocols under varying shortage levels, we visualized the survival-capacity curve (SCC) and allocation-capacity curve (ACC). Our experiments were conducted on firewall-protected servers. The training time for each capacity-specific model was less than 30 minutes using a GPU backend. For more technical details in hyperparameter selections and model configurations, please refer to appendix A.2.

## 5.3 BASELINE PROTOCOLS

Our proposed method was compared with the following existing triage protocols: **Lottery**: Ventilators are randomly assigned to patients who are in need. **Youngest First**: The highest priority is given to the youngest patients. **SOFA**: Patients' prioritization is discretized into three levels (0-7: high, 8-11: medium, and 11+: low) with the lottery serving as the tiebreaker. **Multiprinciple (MP)**: Each patient

Table 3: Impact of triage protocols on survival, fairness, and allocation rates with limited ventilators (B = 40) corresponding to about 50% scarcity. TDDQN and TDDQN-fair are without and with fairness rewards, respectively. Fairness metric is demographic parity ratio (DPR, the ratio between the smallest and the largest allocation rate across patient groups, 100% indicating non-discriminative). Standard deviations are from 10 experiments with different seeds. We bold the ethnoracial group results with the highest survival rate, DPR, and overall allocation rates. We underline the metrics that fall within one standard deviation of the best result.

| | Performance | Fairness | Allocation Rates | | | | |
|---|---|---|---|---|---|---|---|
| | Survival, % | DPR, % | Overall, % | Asian, % | Black, % | Hispanic, % | White, % |
| **Lottery** | 75.17 ± 0.45 | **96.89 ± 1.55** | 75.60 ± 0.28 | 74.95 ± 1.81 | 75.65 ± 1.10 | 76.08 ± 0.90 | 75.68 ± 0.33 |
| **Youngest** | 77.24 ± 0.07 | 86.39 ± 0.27 | 75.65 ± 0.09 | 75.59 ± 0.50 | 81.73 ± 0.31 | 84.44 ± 0.30 | 72.94 ± 0.12 |
| **SOFA** | 80.88 ± 0.32 | 92.37 ± 1.01 | 77.92 ± 0.25 | 75.06 ± 1.44 | 73.58 ± 0.77 | 78.66 ± 1.24 | 79.10 ± 0.45 |
| **MP** | 81.99 ± 0.14 | 90.68 ± 0.95 | 78.34 ± 0.20 | 77.91 ± 1.48 | 74.33 ± 0.66 | 81.96 ± 0.56 | 78.81 ± 0.34 |
| **TDDQN** | 84.76 ± 0.24 | 86.91 ± 3.45 | 81.80 ± 0.26 | 78.03 ± 0.99 | 72.48 ± 0.74 | 81.71 ± 0.44 | 84.45 ± 0.22 |
| **TDDQN-fair** | **85.41 ± 0.23** | 95.24 ± 1.65 | **81.90 ± 0.24** | 80.01 ± 1.79 | 79.95 ± 1.05 | 81.26 ± 1.24 | 82.80 ± 0.42 |

is assigned a priority point based on their SOFA score (0-8: 1, 9-11: 2, 12-14: 3, 14+: 4). Patients with severe comorbidities receive an additional 3 points. In case of ties, priority is given to patients in a younger age group (Age groups: 0–49, 50–69, 70–84, and 85+). If ties still exist, a lottery is conducted to determine the final allocation.

## 5.4 RESULTS

Our findings regarding the survival rates for different triage protocols under various levels of ventilator shortage are illustrated in Figure 3. Across all triage protocols, higher ventilator capacities are associated with increased allocation rates, thereby resulting in saving more lives. The Area Under the Survival-Capacity Curve (AUSCC) serves as an indicator of the overall performance in terms of life-saving abilities under different levels of ventilator shortage. Similarly, the Area Under the Allocation-Capacity Curve (AUACC) reflects the overall performance of ventilator utilization rates under varying levels of shortage. In Panel A of Figure 3, our methods TDDQN and TDDQN-fair exhibit higher AUSCC values compared to all other baseline protocols, demonstrating the superiority of our models in terms of life-saving efficacy. Likewise, in Panel B, TDDQN and TDDQN-fair demonstrate higher AUACC values compared to other baselines, indicating their superiority in terms of ventilator utilization.

Figure 4 showcases the allocation-capacity curves for different ethnoracial groups and triage protocols in Panels A through E. Panel I displays the corresponding AUACC values. Only Lottery and our TDDQN-fair model exhibit minimal disparities, but TDDQN-fair surpasses Lottery in terms of allocation rates and life-saving efficacy. Conversely, all other triage protocols display a preference for specific ethnoracial groups. For example, Youngest and MP favor the Hispanic group, while SOFA and TDQN favor the White group.

Table 2: Ablation study on rewards

| | Rewards | Survival, % | DPR, % |
|---|---|---|---|
| TDDQN - fair | $R_v + R_t + R_f$ | 85.41 | 95.24 |
| TDDQN | $R_v + R_t$ | 84.76 | 90.68 |
| TDDQN-$R_v$ | $R_t + R_f$ | 82.60 | 94.47 |
| TDDQN-$R_t$ | $R_v + R_f$ | 76.02 | 95.32 |

We conducted a detailed analysis in Table 3 under the scenario where approximately 50% of ventilators are unavailable. Our results show that TDDQN-fair achieves the highest survival rate and allocation rate among the triage protocols. It ranks second in terms of demographic parity ratio, following closely behind the Lottery protocol. These findings confirm the effectiveness of our TDDQN-fair approach in improving both survival rates and fairness simultaneously. Importantly, the inclusion of fairness rewards does not compromise its life-saving capabilities compared to the TDDQN configuration.

Our proposed model also demonstrates enhanced fairness and life-saving outcomes in the settings where daily reassessment is taken into account, and the shortage of ventilators does not lead to immediate patient fatalities. For detailed results, please refer to sections A.3 and A.4.

## 5.5 ABLATION STUDIES

We conducted ablation studies to analyze the impact of each reward component. The results indicate that if the ventilation cost or terminal conditions are ignored, the model can achieve a fair distribution,

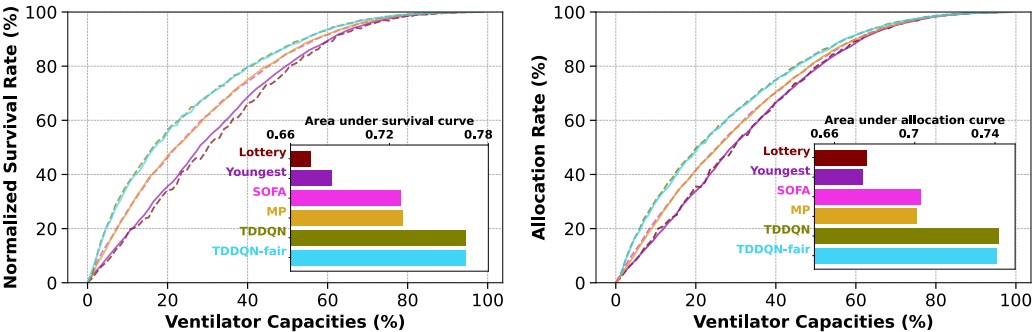

Figure 3: Impact of triage protocols on survival rates and allocation rates under varying levels of ventilator shortages. The maximum daily demand for ventilators in the testing set is considered as full capacity (100%). We scale the number of survivors to a range of [0, 100%] to represent the survival rate. Any capacity below full capacity results in lower survival rates due to ventilator shortages. The allocation rate is calculated by dividing the total number of ventilators allocated by the total number of the ventilators requested. The bar plot associated with each panel indicates the area under the survival-capacity curve and allocation-capactiy curve, respectively, where a larger value indicates that the protocol can save more lives across different levels of shortages. Notably, the MP and SOFA curves exhibit overlap, indicating similar allocation patterns. Similarly, the lottery and youngest curves show close proximity, as do our two TDDQN configurations.

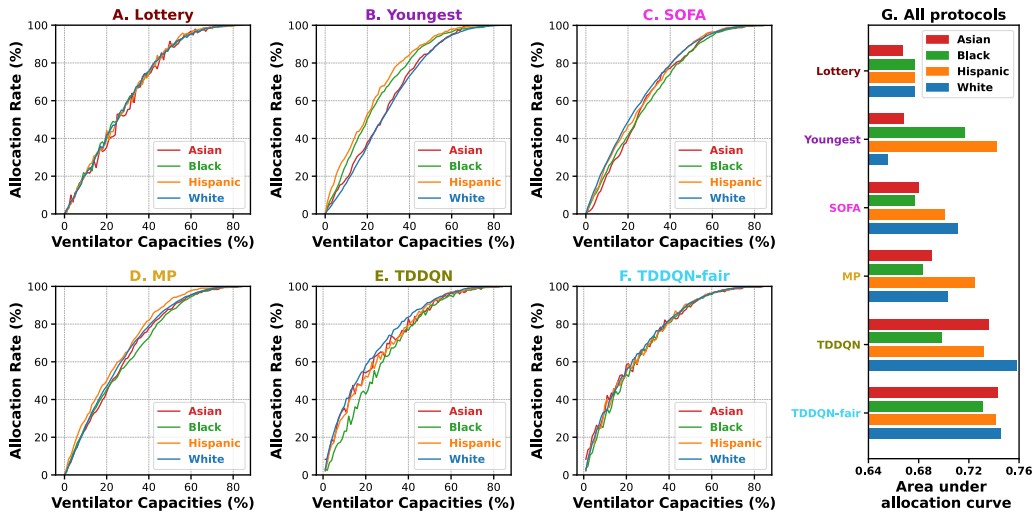

Figure 4: Allocation Rates across different protocols and ethnoracial groups. Panels A to F depict the ethnoracial differences in allocation rates when applying different protocols. Each panel focuses on a specific protocol and shows how the allocation rates vary across ethnoracial groups. Panel G presents the area under the allocation-capacity curve across all protocols and all ethnoracial groups.

but at the expense of reduced life-saving ability. This finding underscores the superiority of our proposed model in achieving both optimized and equitable allocation. For additional ablation studies, please refer to appendix A.5.

## 6 SUMMARY

In this study, we developed a transformer-based deep Q-network to integrate disease progression of individual patients and the interaction effects among patients in order to optimize for an efficient and fair allocation policy. Our proposed model outperforms existing protocols used by different

state governments in the U.S., by saving more lives and achieving a more equitable distribution of ventilators, as demonstrated by experiments on real-world data. We also showed that adding the fairness component into the reward of our multi-objective TDDQN improved fairness to an ideal level while still maintaining superior overall patient survival. We refer the readers to Appendix 5 for a detailed discussion on the limitations and future directions of this study.

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

## A    APPENDIX

We organize the appendix as follows. Section A.1 provides descriptive statistics of the dataset used in this study. In section A.2, we provide additional details on the hyperparameters and model framework employed in our experiments. Section A.3 presents the results of the setting when daily reassessment for ventilator allocation is taken into account. In Section A.4, we discuss the scenario in which the lack of allocation of ventilators does not lead to immediate fatality. In section A.5, we include further ablation studies, in addition to the reward shaping discussed in the main text. Section A.6 summarizes the limitations of this study and outlines potential future directions. Furthermore, we have created separate files containing the codebase for this study in the supplemental materials, which will also be released on GitHub upon acceptance. We will share the de-identified dataset upon publication of the paper.

A.1 DATASET

Table A1: Summary descriptive statistics of the variables in our dataset. (N = 11773). Binary variables are presented using positive numbers and percentages, while continuous and ordinal variables are described using means and standard deviations.

| Variables | N / Mean | % / Std |
|---|---|---|
| **Demographics** | | |
| Sex | | |
|    Female | 4630 | 39.30% |
|    Male | 7143 | 60.70% |
| Race/Ethnicity | | |
|    Asian | 479 | 4.07% |
|    Black | 1736 | 14.75% |
|    Hispanic | 1274 | 10.82% |
|    White | 7509 | 63.78% |
| Age (yr) | 62 | 15.49 |
| **Vital Signs** | | |
| Pulse (BPM) | 90.3 | 22.18 |
| SpO2 (%) | 96.9 | 5.03 |
| Respirations (BPM) | 21.2 | 8.15 |
| BMI (kg/m²) | 29.6 | 8.74 |
| Systolic BP (mmHg) | 122.1 | 27.64 |
| Diastolic BP (mmHg) | 66.7 | 18.64 |
| Temperature (°F) | 97.8 | 1.94 |
| **SOFA scores** | | |
| Respiratory | 1.9 | 1.17 |
| Coagulation | 0.7 | 0.97 |
| Hepatic | 0.5 | 0.88 |
| Cardiovascular | 2.7 | 1.40 |
| Neurological | 2.5 | 1.43 |
| Renal | 0.9 | 1.18 |
| **Comorbidities** | | |
| Acute Myocardial Infarction | 2622 | 22.27% |
| Congestive Heart Failure | 4716 | 40.06% |
| Peripheral Vascular Disease | 3387 | 28.77% |
| Cerebrovascular Disease | 3350 | 28.45% |
| Dementia | 577 | 4.90% |
| Chronic Obstructive Pulmonary Disease | 3673 | 31.20% |
| Rheumatic Disease | 525 | 4.46% |
| Peptic Ulcer Disease | 596 | 5.06% |
| Mild Liver Disease | 1855 | 15.76% |
| Diabetes | 4132 | 35.10% |
| Diabetes with Wound Complications | 2391 | 20.31% |
| Hemiplegia or Paraplegia | 1091 | 9.27% |
| Renal Disease | 3673 | 31.20% |
| Cancer | 2260 | 19.20% |
| Moderate or Severe Liver Disease | 872 | 7.41% |
| Metastatic Cancer | 1173 | 9.96% |
| AIDS | 75 | 0.64% |
| COVID-19 | 1436 | 12.20% |
| **Outcome** | | |
| Death | 2980 | 25.31% |

## A.2   HYPER-PARAMETERS

Table A2: Hyperparameters for model training details, TDDQN framwork and the markov decision process in the study formulation

| Hyper-parameters | |
|---|---|
| **Model training** | |
| Batch size | 32 |
| Learning rate | 3e-5 |
| Steps per update | 500 |
| Training step | 50000 |
| **TDDQN** | |
| Hidden size | 1024 |
| Embedding dim | 1024 |
| Attention head | 16 |
| Max seq length | capactiy + 24 |
| Double Q networks sync frequency | 500 |
| **MDP** | |
| Scalar for ventilator cost ($c_1$) | -0.1 |
| Scalar for fairness penalty ($\lambda$) | 1000 |
| Discount factor | 0.95 |
| Reply buffer size | 16000 |

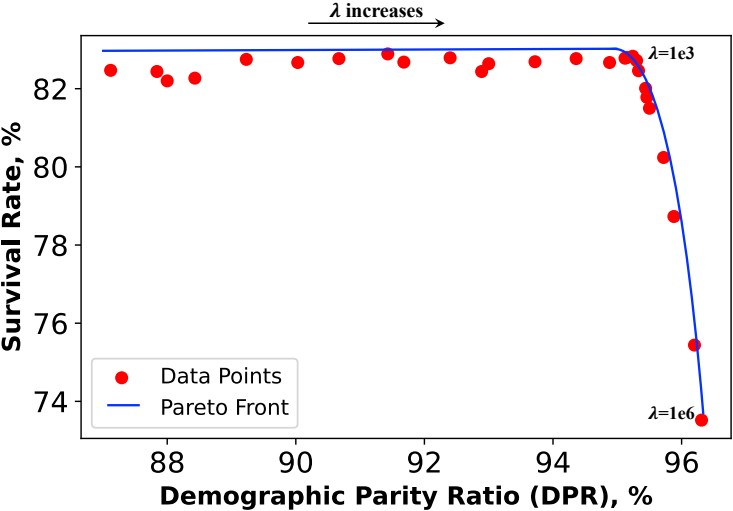

Figure A1: The survival-fairness Pareto frontier was examined under a 50% shortage of ventilators. Each data point corresponds to the outcomes from distinct values of $\lambda$, which balance the trade-off between allocation effectiveness and fairness. As $\lambda$ increases (red dots move to the right), fairness can be enhanced until a turning point is reached, at which the survival rate begins to decline. If $\lambda$ is increased to an exceedingly large value (e.g., 1e6), the model will tend to favor a protocol that prioritizes fairness without due consideration for life-saving. We reported the results from the turning point ($\lambda = 1e3$), where the model is enhanced in fairness without compromising the survival rate.

A.3 ALLOCATION WITH DAILY REASSESSMENT

In the main text, we discuss the circumstances where patients should not have their ventilators withdrawn once allocated until the end of their ICU course due to the scarcity of resources. However, in real-world settings, there may be situations where the withdrawal of mechanical ventilation from one patient to give it to another becomes necessary during a ventilator shortage, particularly when a patient with higher priority requires the ventilator. Piscitello et al. (2020) summarized that 22 guidelines in the United States provide instructions on when and what criteria should be used to reassess the priority of a patient. However, a consensus has not been reached on this matter. In the main results, we take a conservative approach by emphasizing that the allocation of ventilation resources from one patient to another should be determined solely by physicians, considering the potential ethical concerns involved. Here, we present the results of our experiment, which allows for daily reassessment, meaning that ventilators are allocated each day based on predefined protocols. Patients who have been already receiving ventilation do not receive priority solely based on their current possession of ventilators. This experiment serves as evidence that our proposed methods also work effectively when reassessment is taken into account, which provide a comprehensive picture across the spectrum connecting both extremes. In real clinical practice, physicians and stakeholders can adjust the frequency and criteria for reassessment based on their expertise and judgment.

Similarly to the settings without reassessment, we present our results in Table A3, Figure A2, and Figure A3. We find that our proposed TDDQN-fair method improves the survival rate compared to all baseline protocols and achieves fair allocation across ethnoracial groups. The TDDQN-fair configuration, with fairness rewards, yields higher survival rates than TDDQN alone and results in a more equitable distribution (with the second-highest DPR, only smaller than the lottery protocol). Furthermore, when considering reassessment, the survival rates of all protocols are higher than when applied them in the non-reassessment setting. This suggests that withdrawing ventilators and reallocating them to patients with higher priority can save more lives. However, ethical considerations should be taken into account, and the decision of whether to include reassessment should be left to the physicians and stakeholders.

Table A3: Impact of triage protocols on survival, fairness, and allocation rates with limited ventilators (B = 40) corresponding to about 50% scarcity, when daily reassessment are considered. TDDQN and TDDQN-fair are without and with fairness rewards, respectively. Fairness metric is demographic parity ratio (DPR, the ratio between the smallest and the largest allocation rate across patient groups, 100% indicating non-discriminative). Standard deviations are from 10 experiments with different seeds. We bold the ethnoracial group results with the highest survival rate, DPR, and overall allocation rates. We underline the metrics that fall within one standard deviation of the best result.

| | Performance | Fairness | Allocation Rates | | | | |
|---|---|---|---|---|---|---|---|
| | Survivals | DPR, % | Overall, % | Asian, % | Black, % | Hispanic, % | White, % |
| **Lottery** | 83.11 ± 0.40 | **98.95 ± 0.54** | 94.37 ± 0.11 | 94.74 ± 0.79 | 94.32 ± 0.34 | 94.20 ± 0.55 | 94.41 ± 0.12 |
| **Youngest** | 81.38 ± 0.10 | 94.63 ± 0.12 | 93.47 ± 0.10 | 92.72 ± 0.17 | 96.08 ± 0.05 | 97.18 ± 0.03 | 91.96 ± 0.02 |
| **SOFA** | 87.88 ± 0.30 | 98.02 ± 0.63 | 94.38 ± 0.58 | 92.96 ± 0.60 | 93.83 ± 0.15 | 94.50 ± 0.35 | 94.77 ± 0.13 |
| **MP** | 88.46 ± 0.13 | 96.51 ± 0.50 | 94.41 ± 0.29 | 92.05 ± 0.49 | 93.70 ± 0.20 | 95.37 ± 0.14 | 94.63 ± 0.05 |
| **TDDQN** | 90.44 ± 0.19 | 95.08 ± 0.68 | 95.18 ± 0.76 | 94.05 ± 1.22 | 94.66 ± 1.75 | 92.73 ± 1.35 | 96.02 ± 0.36 |
| **TDDQN-fair** | **90.53 ± 0.17** | 98.51 ± 0.61 | **95.25 ± 0.72** | 94.42 ± 0.81 | 95.21 ± 0.57 | 94.87 ± 0.57 | 95.49 ± 0.81 |

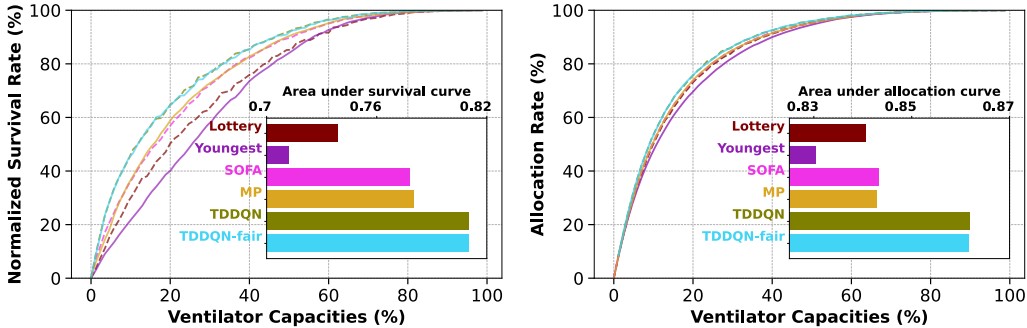

Figure A2: Impact of triage protocols on survival rates and allocation rates under varying levels of ventilator shortages, when daily reassement is considered. The maximum daily demand for ventilators in the testing set is considered as full capacity (100%). We scale the number of survivors to a range of [0, 100%] to represent the survival rate. Any capacity below full capacity results in lower survival rates due to ventilator shortages. The allocation rate is calculated by dividing the total number of ventilators allocated by the total number of the ventilators requested. The bar plot associated with each panel indicates the area under the survival-capacity curve and allocation-capactiy curve, respectively, where a larger value indicates that the protocol can save more lives across different levels of shortages. Notably, the MP and SOFA curves exhibit overlap, indicating similar allocation patterns. Similarly, the lottery and youngest curves show close proximity, as do our two TDDQN configurations.

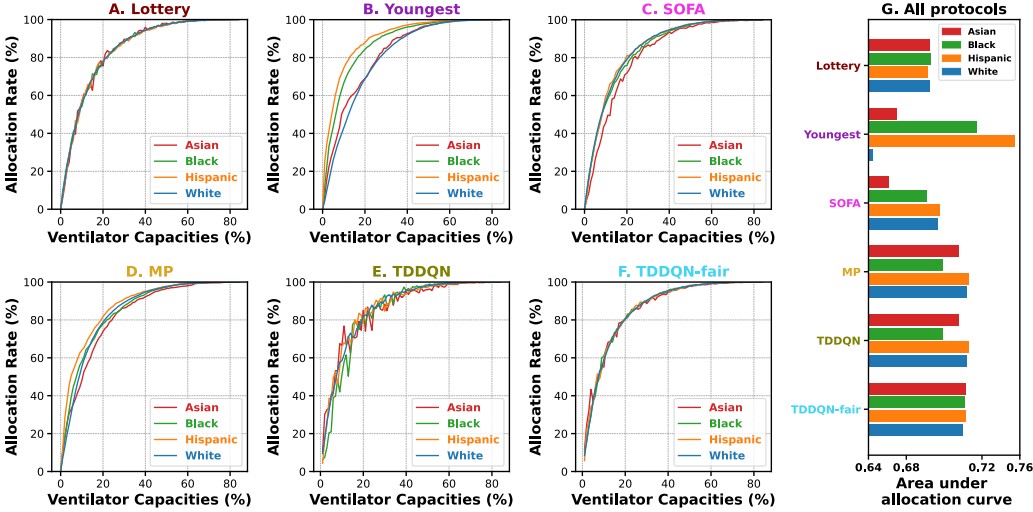

Figure A3: Allocation Rates across different protocols and ethnoracial groups when daily reassessment is considered. Panels A to F depict the ethnoracial differences in allocation rates when applying different protocols. Each panel focuses on a specific protocol and shows how the allocation rates vary across ethnoracial groups. Panel G presents the area under the allocation-capacity curve across all protocols and all ethnoracial groups.

## A.4 MORE EXPERIMENTS

In the main text, we addressed a scenario in which patients die immediately if they request ventilators, but cannot be allocated due to resource shortages. This assumption was grounded in the understanding that ventilators are life-saving devices, and patients in critical care units are critically ill. Additionally, we are unable to obtain or simulate patient trajectories when a patient needs ventilators but is unable to receive one, as collecting this type of data is impractical. Thus, it is impossible to precisely estimate which patients or how many will die due to the lack of ventilation.

In this section, we make an effort to approximate the probability of mortality resulting from a lack of ventilation using a numerical value between 0 and 1. It's important to emphasize that these experiments should only be interpreted as a means to validate the generalizability of our proposed models. The exact probability can only be discerned by healthcare professionals and may vary depending on various factors such as healthcare resource availability, pandemic stages, seasonal variations, geographical locations, and other contextual considerations.

The results presented in Table A4 clearly demonstrate that our proposed model surpasses all other protocols in terms of survival rates. It also achieves fair allocations that are on par with the lottery protocol and significantly outperforms the other three protocols. These superior outcomes hold true across various levels of the probability of death due to ventilation shortages, validating the versatility and applicability of our proposed model across various levels of urgency in healthcare resources allocation.

Table A4: Impact of triage protocols on survival and fairness allocation rates with limited ventilators (B = 40) corresponding to about 50% scarcity, where unmet ventilator requests do not result in immediate fatalities. The probability denotes an estimated rate of early patient deaths attributed to unmet ventilator needs. We scale the number of survivors to a range of [0, 100%] to represent the survival rate. Fairness metric is demographic parity ratio (DPR, the ratio between the smallest and the largest allocation rate across patient groups, 100% indicating non-discriminative). We bold the protocols with the highest survival rate and DPR at different levels of probability. We underline the metrics that fall within one standard deviation of the best result.

| Probability of Death due to No Ventilation | Lottery | | Youngest | | SOFA | | MP | | TDDQN-fair | |
|---|---|---|---|---|---|---|---|---|---|---|
| | Survival | DPR | Survival | DPR | Survival | DPR | Survival | DPR | Survival | DPR |
| 1 | 82.86 | **98.98** | 81.34 | 94.65 | 87.74 | 98.04 | 88.40 | 96.46 | **90.54** | 98.54 |
| 0.9 | 83.60 | **98.86** | 82.29 | 94.36 | 88.13 | 97.86 | 88.83 | 96.43 | **90.80** | 98.42 |
| 0.8 | 84.14 | **98.77** | 83.10 | 93.99 | 88.66 | 97.73 | 89.35 | 96.42 | **91.47** | 98.40 |
| 0.7 | 84.87 | **98.72** | 84.25 | 93.52 | 89.11 | 97.66 | 89.82 | 96.19 | **91.83** | 98.36 |
| 0.6 | 85.38 | **98.63** | 85.19 | 93.02 | 89.73 | 97.63 | 90.27 | 96.06 | **92.04** | 98.24 |
| 0.5 | 86.37 | **98.52** | 86.35 | 92.55 | 90.50 | 97.58 | 90.87 | 95.95 | **92.44** | 98.23 |
| 0.4 | 87.65 | **98.47** | 87.99 | 91.73 | 91.31 | 96.74 | 91.80 | 95.22 | **93.72** | 98.23 |
| 0.3 | 89.13 | 98.04 | 89.76 | 90.46 | 92.32 | 96.46 | 92.77 | 94.86 | **94.15** | **98.18** |
| 0.2 | 91.02 | 97.87 | 91.69 | 88.93 | 93.84 | 95.60 | 94.18 | 94.21 | **95.42** | **98.14** |
| 0.1 | 94.27 | 97.49 | 94.80 | 87.19 | 95.91 | 94.12 | 96.12 | 94.00 | **96.36** | **98.10** |

## A.5 MORE ABLATION STUDIES AND BASELINES

In this section, we report additional ablation studies and baselines besides the ablation experiments on rewards in the main text. See Table A5 for survival and fairness results with limited ventilators (B = 40) corresponding to about 50% scarcity.

**DDQN-individual** In this experiment, our focus is on modeling the Markov decision process for individual patients using a double DQN network. Unlike the TDDQN model, which considers the dynamic needs of all patients in the ICU unit at a given time, we specifically examine the trajectory progression of a single patient. Similarly to other baseline approaches, we assign a priority score to each patient based on the expected cumulative rewards when allocating the ventilator to them. However, since our emphasis is on individual patients, the group fairness reward cannot be taken into account in this particular experiment. The result indicates that this method has limited effectiveness in improving the survival rate compared to the baseline protocol. Additionally, it also cannot ensure equitable allocation.

**TDQN-fair** Our TDDQN-fair model addresses the issue of overestimation of action values and stabilizes the training processes by leveraging the double deep Q network. However, in this specific experiment, we use a single neural network that simultaneously estimates both the current and target action values. This configuration achieves comparable results to TDDQN-fair; however, it exhibits larger fluctuations in both survival and fairness, as indicated by a larger standard deviation.

**TDDQN-lottery** In the training of the TDDQN-fair model, we collect a fixed-length set of trajectories in advance and stored them in a replay buffer. These trajectories are obtained by deploying our proposed models for ventilator allocation. We iteratively repeat the process of training the model and collecting new trajectories using the deployed model. For this experiment, we collect the trajectories specifically when using the lottery protocol. It's important to note that the data collection process was executed only once at the beginning, and the model was subsequently trained on this pre-collected experience. Since the lottery protocol inherently promotes a fair allocation, the actions leading to inequitable allocations are not penalized adequately. This experiment yields similar survival rates but less equity compared to our TDDQN-fair model.

**TDDQN-early** This experiment is designed to simulate the scenario that occurred at the initial stage of the pandemic in the real world, where only a limited amount of allocation experience data was available. Unlike in the training of TDDQN-fair, where we sampled the training data from the patients' trajectories spanning the first 16 months of the pandemic, this experiment focuses on sampling data solely from the first 3 months of the pandemic (03/15/2020 - 06/14/2020). Despite reducing the training set to approximately 20%, this configuration continues to outperform all baseline protocols in terms of survival rates and promotes an equitable distribution of ventilators.

**Grand-Clément et al. (2021)** introduced a data-driven decision-tree-based approach for optimizing ventilator allocation. Their allocation protocol factors in the BMI, age, and SOFA score of patients, classifying them into two priority levels: high and low. Admission time is employed as a tie-breaker within the same priority group. It is clear that our proposed model surpasses this data-driven baseline in terms of both effectiveness and fairness.

Table A5: Ablation studies.

| Model | Survival, % | DPR, % |
|---|---|---|
| TDDQN-fair | **85.41 ± 0.23** | **95.24 ± 1.65** |
| DDQN-individual | 83.39 ± 0.17 | 88.91 ± 0.84 |
| TDQN-fair | 84.32 ± 0.53 | 94.54 ± 2.27 |
| TDDQN-lottery | 85.02 ± 0.29 | 92.73 ± 2.01 |
| TDDQN-early | 83.70 ± 0.19 | 94.58 ± 1.24 |
| Grand-Clément et al. (2021) | 76.16 ± 0.70 | 92.24 ± 1.97 |

## A.6 LIMITATIONS

In this study, we acknowledge several limitations that can serve as valuable directions for advancing our research. Firstly, our model was developed using data solely from one health system. Although we carefully divided the data into training, testing, and validation sets, utilizing both simulator data and real-world data, the generalizability of our findings may be limited. To address this, it would be beneficial to obtain data from multiple sites, allowing us to assess and discuss the model's performance across different healthcare settings. Additionally, our current model requires separate training for each specific ventilator shortage level. We aim to overcome this limitation by developing a more generalized model that can adapt with minimal changes as the capacity fluctuates. This would enhance the model's flexibility and practicality in real-world scenarios. Lastly, our model has the potential to extend its application to other critical healthcare resources such as ECMO (Extracorporeal Membrane Oxygenation) and ICU beds. However, due to data limitations, we are constrained to conducting experiments and evaluations on these resources only when data becomes available. Expanding our research to include these resources would provide a more comprehensive understanding of their utilization and optimization. By addressing these limitations and exploring the proposed directions, we can enhance the robustness, applicability, and effectiveness of our study.

