# OpenReview forum: "Deep Reinforcement Learning for Efficient and Fair Allocation of Health Care Resources"
_ICLR.cc/2024/Conference — Submitted to ICLR 2024_

### Official Review · Reviewer_om5H · 2023-10-30

**Soundness:** 2 fair
**Presentation:** 3 good
**Contribution:** 2 fair
**Rating:** 5
**Confidence:** 3

**Summary:**

The paper proposes a deep Q-learning approach for resource prioritization in hospitals. They learn a Q function that given the state of all patients and current ventilation status at each bed, for each vector of the next ventilation decisions (a binary vector) outputs the value per patient. These values can subsequently be utilized to formulate a policy. The method's performance is assessed using real ICU data.

**Strengths:**

- The problem is well-motivated and holds significant importance.
- The presentation is mostly clear.
- Real data is used for evaluation.

**Weaknesses:**

1. I don't think the proposed Q-learning framework fits this problem. In fact, the proposed Q-network cannot see the interaction of patients; changing $a_i$ does not change $Q_\theta(s)_{j,a_j}$, $j \neq i$. Please clarify if this is not the case.
2. It's not clear what value this Q function is estimating. Further elaboration is required here.
3. I cannot see how survival rate as a measure of performance is estimated from offline data. What off-policy evaluation technique is used?
4. Minor typos: Eq. 3: $x'$ to $x'_i$.  Page 3: in 3 to in Fig. 3.

**Questions:**

Please refer to the weaknesses.

---

> ### Author Response · Authors · 2023-11-14
> **Response to Reviewer om5H29**
>
> We appreciate the reviewer's efforts in helping us with improving the manuscript and providing an opportunity to make clarifications. We have corrected the minor typo as suggested. Please find our point-by-point response below.
>
> > 1. I don't think the proposed Q-learning framework fits this problem. In fact, the proposed Q-network cannot see the interaction of patients; changing  $a_i$ does not change $Q_\\theta(s)_{j,a_j}, j\\neq i$ . Please clarify if this is not the case.
>
> We appreciate the reviewer for bringing this out. We would like to provide further clarification regarding patient interactions within our model. Patient interactions are considered in both the states and actions. In terms of states, each patient state $s_i$, is input into the transformer framework as an individual token. Consequently, changing the value of $s_i$ can impact all $Q_\\theta(s)_{j,a_j}, j\\neq i$, due to the attention mechanism within the transformer architecture. This interaction is analogous to how changing a single word in a sentence affects the output embeddings of all other words in the sentence, considering the contextual information. We also conducted experiments (DDQN-individual model in Appendix 5) where interactions were ignored, and changing the value of $s\_i$ did not impact any $Q\_\\theta(s)\_{j,a\_j}, j\\neq i$. This interaction-ignored model exhibited lower performance in terms of both survival rate and fairness.
>
> Regarding actions, the interaction lies in the constraints imposed by the available ventilators, denoted as $B$. Modifying the action $a\_i$ will not affect $Q\_\\theta(s)\_{j,a\_j}, j\\neq i$. However, it does have an influence on the greedy action,  $a^* = \\arg\\max\_{a \\in\\mathcal{A}} Q(s,a)$, which is determined by identifying the extra extra ($B-|I|$)-largest indices of $ \\{  (Q\_\\theta(s)\_{i, 1} -Q\_\\theta(s)\_{i, 0}) | 1\\leq i \\leq N, I\_i=0\\}$. For instance, if we change $a\_i$ from 1 to 0, indicating that a patient's ventilator is freed up, the count of patients already on ventilation ($|I|$ ) decreases by 1,  and the greedy action will be adjusted to find the extra  ($B-|I-1|$)  indices of $ \\{  (Q\_\\theta(s)\_{i, 1} -Q\_\\theta(s)\_{i, 0}) | 1\\leq i \\leq N, I\_i=0\\}$ to allocate ventilators. Therefore, the action $a\_j$ ,  whose $Q\_\\theta(s)\_{j, 1} -Q\_\\theta(s)\_{j, 0}$ ranks exactly at the $B-|I-1|$ position will change from 0 to 1.
>
> > 2. It's not clear what value this Q function is estimating. Further elaboration is required here.
>
> The Q function estimates the expected cumulative reward (Eq. (5)) when allocating ventilators under the constraint of the total available ventilators, taking into account the clinical conditions of the patients. This Q function estimates a trade-off between the survival rate of all patients and the fairness of allocation across different ethnoracial groups. To address computational challenges, we decomposed the overall $Q(s,a)$ into the sum of individual $Q_\\theta(s)_{i,a_i}$ and used an attention mechanism to consider interactions among patients. We have included a clarification of the Q function in Section 4 of the revision.
>
> > 3. I cannot see how survival rate as a measure of performance is estimated from offline data. What off-policy technique is used?
>
> Thank you for your question. We've developed a simulator using real-world data, obviating the need for off-policy evaluation techniques. The simulator allows us to directly evaluate our proposed strategy and compare its performance with existing protocols. The usage of a simulator derived from real-world clinical data also enhances the utility of our proposed model compared to off-policy techniques. Within the simulator, a patient progresses to their subsequent stage in the real clinical trajectory if a ventilator is allocated; otherwise, they experience immediate decease due to ventilator shortage. The death count comprises patient fatalities within their clinical progressions and those resulting from insufficient ventilation. The survival rate is simply calculated as 1 minus the death rate. In addition, we present results from alternative simulators in the appendix, the simulators where patients do not progress to immediate death in the absence of ventilators.

---

> ### Comment · Reviewer_om5H · 2023-11-21
>
> Thank you for your response.
>
> I agree the Q-network captures state interactions. What I can't see is how actions are interacting beyond the immediate budget constraint. Let me put it this way: what is $Q_\theta(s)_{i, a_i}$ estimating? Can you write a Bellman update for this? Is this the expected cumulative return from individual $i$ assuming action $a_i$ is taken? Then why it does not depend on $a_j$? Maybe I'm missing something but I think what decision I make about patient $j$ at this point should affect the expected cumulative return for patient $i$ as well.
>
> Just to give you an idea of what I'm thinking about, consider this simple example. Let there be only 3 patients with a budget $B=1$ ventilators. Consider the following $Q_\theta(s)$.
>
> |      |$a=0$ | $a=1$ |
> | - | - | - |
> | $i=1$ | 0.25 | 0.5  |
> | $i=2$ | -1 | -1  |
> | $i=3$ | -1 | 0.75 |
>
> Let's look at the problem from $i=1$ perspective: For action profile $(0, 0, 1)$, patient 3 most likely survives and needs a ventilator more than me at the next step. So, I'm unlikely to be ventilated. But for action profile $(0, 1, 0)$ both other agents will die and I'm more likely to receive the ventilator next time! Therefore the actions of the others matter for my value but for both action profiles $Q_\theta(s)_{1, 0}$ is the same.
>
> I acknowledge considering all forms of action interactions require a massive Q network. But it is not clear what we are missing with the current definition of Q.
>
> Overall, I have some concerns with the framework of this paper, its presentation, and relevance to ICLR. Therefore, while I appreciate the authors' efforts, I'd like to stick to my current evaluation.

---

> ### Author Response · Authors · 2023-11-21
> **Hi thanks for the response but let's clarify the misunderstanding**
>
> Hello thanks for the response. Your example is great, and we believe we share the same understanding. It is true that TDDQN uses a parameterized Q network which doesn't capture every interaction. However, we would hope to invite you to understand the complexity of the problem.
>
> The original Bellman equation for this problem involves exponentially many actions, which would create a massive network whose input size is **$O(N \\times 2^N)$**. There is no hope to construct such a network and train it in practice. Classical DQN simply doesn't work. In fact, one cannot directly apply any RL algorithm to this problem. This is why we couldn't find any prior work using RL for this problem. In other words, **it is exponentially hard to capture all patients' interactions**. This is what motivates our work.
>
> In this paper we do not try to capture all interactions. Instead, we propose a reparametrization of Q network to circumvent the issue of exponentially many actions: $Q(s,a) \\approx \\sum_{i=1}^N Q_{\\theta}(s)\_{i,a\_i}$.
> **This parametrization reduces the network input dimension to $O(N)$**. In the meantime, this parametrization of Q network also leads to a policy class where greedy policies can be calculated very easily, i.e., choosing actions by ranking values of $Q_{\theta}$. In fact, our policy class has a spirit similar to the so-called "index policy" in queueing theory which is asymptotic optimal in some special cases (see eg. Whittle's index policy for a multi-class queueing system with convex holding costs).
>
> Your concerns come from the fact that we are using this particular Q parametrization to approximately solve the problem. We agree with what you pointed out. Our reduction of complexity does come at a cost: We agree that this parametrization of Q network doesn't capture all patients' interactions, and that $Q_{\theta}(s)_{i,a_i}$ is only a proxy to patient's value but may not be  accurate. But please bear in mind that we are approximately solving an exponentially hard problem using a Q parametrization with linear space complexity. Please also note that all DQN algorithms use parametrized versions of Q network to save computation costs and there is always approximation error.
>
> To justify that this parametrization works, we conducted extensive experiments using the policies learnt by TDDQN. Our Table 2,3 reported performances of our policy compared to other baselines commonly used in the hospital. You can see that our policies work significantly better than all other baselines. In other words, although TDDQN only provides an approximate solution to the Bellman equation using a simplified Q network, it actually can learn a pretty good SOTA policy beating all other baselines.
>
> We really appreciate your comments and discussions and we hope that we could clarify the misunderstanding

---

> > ### Comment · Reviewer_om5H · 2023-11-21
> >
> > Thank you for your response.
> >
> > It's good that we now agree $Q_\theta(s)_{i,a_i}$ is only a proxy.
> >
> > I also agree there is always an approximation error when using a specific Q network parameterization. However, given the richness of the typical networks, at least in my experience, that's not a concern.
> >
> > So, my formulation concerns have multiple facets:
> > - First, no matter how rich your Q representation is, patient interactions beyond the immediate budget constraint cannot be captured. So, I wish to see how important this approximation is. I recommend a bit of theoretical work or clarification so the reader can understand what we are losing here.
> >
> > - Second, I agree the empirical results are interesting. But given that these are based on a simulation environment, and given my unfamiliarity with the baselines considered, I cannot comment on how significant are these results.
> >
> > - Third, I'm not convinced Q learning is the best framework to solve this problem. For example, given the partial knowledge we have about the transition dynamic, why not approach this problem as a model-based RL by learning $q_i$, $\xi$, $p^{on}$, etc., followed by solving dynamic programming accurately? I know this question might be beyond the context of your paper. I'm just saying you need to better motivate Q learning in this context.
> >
> > I'm sorry that I still feel negative about your work. I think you have done something important and certain improvements can make your paper ready for publication in future venues.

---

> > > ### Author Response · Authors · 2023-11-21
> > >
> > > Dynamic programming, which you proposed, is intractable because $|A| = 2^N$. To put it simply, even if transition model is fully known, representing the Q function requires $O(N \times 2^N)$ space in a computer memory, which is not possible to implement. In other words, again, it is not possible computationally to capture all interactions and state dependences.
> > >
> > > The quality of our algorithm is supported by the strong empirical results, presenting the first result of applying RL to this fairness-aware healthcare problem. We agree with you that it would be awesome to have some theoretical justification. However, establishing an approximation bound for this combinatorially hard optimization problem is highly nontrivial. We suspect the problem is PSPACE-hard. Confirming its complexity and approximation bound would be a problem in the space of theoretical computer science. Although it's fascinating to think about this theory question, it is really beyond the scope of our application paper. The goal of our paper is to demo that RL methods can promote fairness while saving lives.
> > >
> > > Thank you for acknowledging lack of familiarity with this healthcare application. The baselines we compared to are real clinical guidelines used in hospitals and by doctors (cited in Section 1 of submission):
> > >
> > > [1] https://www.health.ny.gov/regulations/task_force/reports_publications/docs/ventilator_guidelines.pdf
> > >
> > > [2] Too many patients. . . a framework to guide statewide allocation of scarce mechanical ventilation during disasters. Chest, 155(4): 848–854, 2019.
> > >
> > > [3] Variation in ventilator allocation guidelines by us state during the coronavirus disease 2019 pandemic: a systematic review. JAMA network open, 3(6):e2012606–e2012606, 2020.
> > >
> > > [4] The sofa (sepsis-related organ failure assessment) score to describe organ dysfunction/failure: On behalf of the working group on sepsis related problems of the european society of intensive care medicine (see contributors to the project in the appendix), 1996.

---

> ### Comment · Reviewer_om5H · 2023-11-22
>
> yes sorry, I didn't mean naive dynamic programming. I just meant given that your problem has a known structure, one might learn the few functions you defined and simply obtain a good model of the transition dynamic. Then one can use any off-the-shelf tools for planning or control. This is probably a baseline I would consider. In general, I was thinking that model-based RL might also be a reasonable solution for such a structured problem.
>
> Anyway, I didn't judge your work based on these alternatives. I'm just saying that Q-learning formulation requires further motivation over these methods, specifically because this is a resource allocation problem and not a typical RL setting.
>
> ps. I slightly increased my score based on the conversation we had.

---

> > ### Author Response · Authors · 2023-11-22
> >
> > Thanks for the discussion. If you know of any off-the-shelf planner that can solve the DP problem with exponentially many actions, please do let us know. We searched for such a planner extensively but couldn't find anything useful. Without such a planner, model-based RL wouldn't work. We still strongly believe that the decomposed Q network and the corresponding index policy provides a most straightforward/practical approximation to utilize the problem structure.
> >
> > Indeed, there could be alternative solutions, but so far we haven't found any.

---

### Official Review · Reviewer_GCVq · 2023-10-31

**Soundness:** 4 excellent
**Presentation:** 4 excellent
**Contribution:** 4 excellent
**Rating:** 10
**Confidence:** 4

**Summary:**

There are many situations within healthcare in which resources such as ventilators are scarce and require the use of carefully thought-out protocols to determine the proper allocation. However, currently there are many different protocols for to handle this allocation which involve conflicting heuristics. Since these decisions are usually sequential, the authors propose using reinforcement learning for
to construct a fair and effective resource allocation protocol. Specifically, they use a transformer-based deep Q-network to integrate patients' disease progressions and interactions during resource allocation. Their experiments show that their method results in both fair and effective allocation of resources.

**Strengths:**

The authors did an excellent job detailing the design of the RL problem and the experimental setups.

**Weaknesses:**

It would have been nice to see the limitations in the main text rather than the appendix. However, it is understandable due to the page limit.

**Questions:**

Is it possible for the authors to disclose the names of the datasets used in this study if they are publicly available?

---

> ### Author Response · Authors · 2023-11-10
> **Response to Reviewer GCVq**
>
> We thank the reviewers for their dedicated effort and valuable time spent on reviewing our manuscript, which has greatly contributed to enhancing the quality of our study. In the upcoming camera-ready version, we will incorporate a summary of the study's limitations to the main text. We will share the de-identified dataset upon publication of the paper.

---

> ### Comment · Reviewer_GCVq · 2023-12-02
> **Response to Authors**
>
> Thank you for the response. I look forward to inspecting the de-identified dataset. I am keeping my score where it stands currently.

---

### Official Review · Reviewer_dm6i · 2023-10-31

**Soundness:** 2 fair
**Presentation:** 2 fair
**Contribution:** 2 fair
**Rating:** 5
**Confidence:** 3

**Summary:**

This paper proposed a transformer-based deep Q-network method for efficient and fair allocation of ventilators in critical care settings. The experiments on a real-world dataset demonstrated that the proposed method achieved both higher patient survival rates and more equitable allocations across different ethnic groups compared to existing heuristic-based policies utilized by different governments.

**Strengths:**

**Originality:** The novelty of this paper mainly lies in the application aspect, in that 1) fairness objectives are incorporated into a DRL framework for healthcare resources allocation modeling 2) the individual patient disease progression and the interaction effects among patients are accounted for simultaneously by utilizing a transformer based Q-network.

**Quality:** The quality of this paper is fair. The experiments showed that the proposed method outperformed existing heuristic-based policies utilized by governments, but did not compare with any data-driven or machine learning baselines. The ablation study showed the effectiveness of different components (the ventilator cost, the patient survival and the fairness in allocation) in the reward function.

**Clarity:** Overall the paper is clear, but the notations are sometimes confusing and more technical details are needed for a better understanding of the method / experiments.

**Significance:** Healthcare resource allocation is an important topic in critical care medicine. A more efficient and fair policy than existing government policies will result in more people being saved at lower levels of costs with minimal disparities among different ethnic groups. Thus this work has high clinical relevance and significance.

**Weaknesses:**

1. The notations are confusing in Section 3.3 and 3.4. Based on the context, are $x_i$ and $s_i$ the same (the medical condition of patient i), and $P^\text{on}$ and $P^\text{vent}$ the same (the ventilation transition)? If so, please ensure that the notations are consistent. Also, I think the action $a$ is N-dimensional and the i-th coordinate $a_i \in \\{0, 1\\}$ is the action applied on patient i, if this is the case, does $I'_i = a$ mean $I'_i = a_i$ in 1? Also, what is the dimension of the overall transition matrix $P$?

2. Currently, the state and action are the concatenations of the medical/ventilator states and the ventilator assignments of individual patients. Will there be scalability issue when the number of patients $N$ is really large and if there is, how to handle that with the current framework?

3. Some details in the transition model are missing. How is $P^\text{vent} (x'_i | x_i)$ determined? From the description of the simulator in Section 5.2, it seems that you consider the factual data, e.g. $P(x'_i | x_i) = 1$ if $x'_i$ is the actual next state for patient i and $P(x'_i | x_i) = 0$ otherwise? Did you consider any counterfactuals? Will the results be negatively impacted by any potential selection bias in the data if only factual data is used? Also, how are $q_i(s)$ (bed assignment distribution) and $\xi$ (initial medical condition distribution) determined?

4. Some details in the Method and Experiments are missing. The learning objective is missing. From the context, I am assuming that you are using Q-learning with the proposed reward and then used the greedy policy with the learned Q function, but it's not clear from what's written now in Section 4. Also, the choice of the ventilator cost $c_1$ and the $\lambda$ to trade-off the fairness reward term are missing in the main paper (found them in appendix). Is there any justification on how they are determined? I am also wondering how the results will change if $c_1$ and $\lambda$ are chosen differently.

**Questions:**

1. Will the dataset be made public if the paper is published? So that the results can be reproduced.

2. The experiments showed that the proposed method outperformed existing government policies. I am wondering if there is any data-driven baselines in the literature and how will your method perform compared to them?
3. There are some confusions in the writing, e.g.

- Page 3, "... proposed model in 1" -> "... proposed model in Figure 1"?
- Page 6: "... the action set $\mathcal{A}$ are defined in Eq. equation 2" -> "... the action set $\mathcal{A}$ are defined in Eq. 2"

**Details Of Ethics Concerns:**

The dataset used seems to be proprietary and the authors did not provide detailed information on the dataset or whether it can be made public upon paper publication.

---

> ### Author Response · Authors · 2023-11-22
> **Response to Reviewer dm6i**
>
> We appreciate the reviewers' efforts in helping us polish the study. Please find our itemized responses to the weaknesses, questions, and ethical concerns below.
>
> > **Weakness 1** The notations are confusing in Section 3.3 and 3.4. Based on the context, are $x_i$ and $s_i$ the same (the medical condition of patient $i$), and $P^{\text{vent}}$ and $P^{\text{on}}$ the same (the ventilation transition)? If so, please ensure that the notations are consistent. Also, I think the action $a$ is N-dimensional and the i-th coordinate $a_i\in\{0,1\}$ is the action applied on patient $i$, if this is the case, does $I_i'=a$ mean $I_i'=a_i$ in 1? Also, what is the dimension of the overall transition matrix $P$?
>
> We thank the reviewer for pointing out the notation inconsistency. We have addressed them in Section 3 of the revision. Specifically, the notation $x_i$ and $s_i$ are the same, we updated all notation of $s_i$ to $x_i$; The notation $P^\text{vent}$ and $P^\text{on}$ are different, meaning the transition of patient ventilation status and medical status, respectively; Typo of $a_i$ is corrected; The dimension of the transition matrix $P$ is $(|S|\times |A|) \times |S|$, where $|S| = Nk+N$ denotes the dimension of the state space, and $|A| = 2^N$ denotes the dimension of the action space. When considering fairness, $|S|$ expands to $Nk + N + 8$ by adding 8 dimensions to account for cumulative total and ventilated patients across four ethnoracial groups. We updated Eq. (5) for the correct formula of $P$.
>
> > **Weakness 2** Currently, the state and action are the concatenations of the medical/ventilator states and the ventilator assignments of individual patients. Will there be scalability issue when the number of patients  is really large and if there is, how to handle that with the current framework?
>
> We thank the reviewer for asking this important question. Naively concatenating all patients' state vectors together does indeed pose computational challenges as the size of the input is exponentially large. To overcome this computational intractability, we therefore proposed the use of a transformer structure that treats patients' states as 'tokens' and inputs them into the model rather than simply concatenating them. By employing this parametrization, we were able to decompose the transitions of patients' states, improving scalability while maintaining interactions among patients. Our approach has already substantially reduced the input complexity from $O(N \times 2^N)$ to $O(N)$. In our implementation, we utilized only one layer of the transformer structure with an embedding dimension of 1024 and 16 attention heads, resulting in just 5.3 million trainable parameters. Our model can recommend resource allocation for over 10,000 beds with a single 32 GB Tesla V100 GPU card. As a reference, the state of Illinois has 3,535 critical care beds [1].
>
> [1] UIC School of Public Health. (2023). Number of hospital beds in Illinois. Retrieved November 21, 2023.
>
> > **"...but did not compare with any data-driven or machine learning baselines."**
>
> Note that the action space is exponential in $N$. It means that even finding the greedy action for a given state is combinatorially hard. Further, even representing the Q function requires $O(N\times 2^N)$ space, which means that any classical method would not be implementable. Given this immense complexity, we didn't find any classical machine learning baseline that can directly apply to our problem immediately.
>
> To circumvent this complexity challenge, we constructed the decomposed Q network approach which not only reduces input size of Q but also leads to a policy class that can be easily computed. We strongly believe that it is a most straightforward and practical approximation to this combinatorially hard problem.

---

> ### Author Response · Authors · 2023-11-22
> **Cont'd Response to Reviewer dm6i**
>
> >  **Weakness 3** Some details in the transition model are missing. From the description of the simulator in Section 5.2, it seems that you consider the factual data, e.g. $P(x'_i| x_i)=1$ if  $x_i$ is the actual next state for patient i and $P(x'_i| x_i)=0$ otherwise? Did you consider any counterfactuals? Will the results be negatively impacted by any potential selection bias in the data if only factual data is used? Also, how are bed assignment distribution $q_i(s)$ and initial medical condition distribution $\xi$ determined?
>
> We thank the reviewer for suggesting the need for clarification. In our project, we built a simulator to model individual patient's state path and the multi-patient ventilator allocation process. The simulator is built on real-world patients data. All patients in our real-world datasets were ventilated. Our simulator reflected scenarios where some patients cannot receive ventilators due to resource shortages. We assumed, following Bhavani et al. [1], that patients not allocated a ventilator do not survive (In our sensitivity analysis, see Appendix A.4, we also conducted additional experiments, exploring the possibility that patients not ventilated may continue to survive along their own trajectory with various probabilities). Thus we believe the simulator can provide a reasonable state-transition model for a random patient. Please note that our TDDQN method can work with any simulator of this kind, and building better simulators to approximate real-world hospital operation will be long-term efforts.
>
> We totally agree that having an accurate state-transition simulation model is crucial for success of deploying ML model in practice, and in particular counterfactuals could be a big concern. In this initial study, we tried our best to collect available data and present a proof-of-concept simulator+RL approach. In the longer term, we do envision this initial efforts would lead to transformative health care resource allocation in real healthcare systems. Thus, beyond the scope of this ICLR submission, we are actively collaborating with physicians and three other academic health systems, each with multiple hospitals to expand the efforts of equitable and effective health care resource allocation.
>
> We clarified these two distributions $q_i(s)$  and  $\xi$  in the revised version of our manuscript, as follows. For the distribution of bed assignments, $q_i(s)$ depends on how new patients are allocated to empty beds. It satisfies $q_i(s) = 0$ if bed $i$ is already occupied, and $\sum_{i=1}^N q_i(s) =  E \sim Pois(\Lambda)$, assuming the number of incoming patients $E$ follows a Poisson distribution with parameter $\Lambda$.  In our implementation, we randomly sampled $E$ patients from the training cohort without replacement and randomly assigned them to the empty beds, where the number of patients $E$ is drawn from a Poisson distribution with $\Lambda=12$, as estimated from our training data. The initial medical condition distribution $\xi$ represents the clinical condition of patients on their first day in the critical care units.
>
> [1] Bhavani et al. "Simulation of ventilator allocation in critically ill patients with COVID-19." American journal of respiratory and critical care medicine 204.10 (2021): 1224-1227.
> > **Weakness 4** Some details in the Method and Experiments are missing. The learning objective is missing. From the context, I am assuming that you are using Q-learning with the proposed reward and then used the greedy policy with the learned Q function, but it's not clear from what's written now in Section 4. Also, the choice of the ventilator cost $c_1$ and the $\lambda$ to trade-off the fairness reward term are missing in the main paper (found them in appendix). Is there any justification on how they are determined? I am also wondering how the results will change if $c_1$ and $\lambda$ are chosen differently.
>
> We used the Q-learning objective with the proposed reward. We incorporated an action constraint to model the limited number of ventilators, when selecting the best action via the greedy policy in Eq.(10) and Eq.(11). We have clarified these details in Section 4 of the revision.  In practice, these parameters should be tuned and determined by doctors. In our experiment, we tested various candidate values of $\lambda$ to identify the best-performing one based on both effectiveness and fairness on the validation set. The $\lambda$ value has an impact on the balance between survival and fairness. In Appendix 2, we present a Pareto frontier obtained by varying the value of $\lambda$. We observe that the increase of $\lambda$ does not compromise the survival rate but enhances fairness until reaching a threshold. When $\lambda$ is set to a very large number, e.g. 1e6, the trained protocol tends towards a random policy, which means equal allocation but lowers the overall survival rate.

---

> ### Author Response · Authors · 2023-11-22
> **Cont'd Response to Reviewer dm6i**
>
> > **Question 1** Will the dataset be made public if the paper is published? So that the results can be reproduced. The dataset used seems to be proprietary and the authors did not provide detailed information on the dataset or whether it can be made public upon paper publication.
>
> We will share the de-identified dataset upon publication of the paper.
>
> > **Question 2** The experiments showed that the proposed method outperformed existing government policies. I am wondering if there is any data-driven baselines in the literature and how will your method perform compared to them?
>
> Note that the action space is exponential in $N$. It means that even finding the greedy action for a given state is combinatorially hard. Further, even representing the Q function requires $O(N\times 2^N)$ space, which means that any classical method would not be implementable. Given this immense complexity, we didn't find any classical machine learning or RL baseline that can directly apply to our problem. To circumvent this complexity challenge, we constructed the decomposed Q network approach which not only reduces input size of Q but also leads to a policy class that can be easily computed. We strongly believe that it is a most straightforward and practical approximation to this combinatorially hard problem.
>
> Further, we discovered a decision-tree heuristic developed for a similar ventilator problem by (Grand-Clement et al. [1]). Although their problem is different from ours, we managed to test this heuristic in our setting and got the following **new results** (added to Appendix A.5). In the comparison, our TDDQN clearly outperforms the tree heuristics.
>
> **Table R1.** Impact of triage protocols on survival, fairness, and allocation rates corresponding to ~ 50% scarcity.
>
> |                      | Performance  | Fairness     | Allocation Rates |              |              |              |              |
> |----------------------|--------------|--------------|------------------|--------------|--------------|--------------|--------------|
> |                      | Survival, %  | DPR, %       | Overall, %       | Asian, %     | Black, %     | Hispanic, %  | White, %     |
> | TDDQN-fair           | 85.41 ± 0.23 | 95.24 ± 1.65 | 81.90 ± 0.24     | 80.01 ± 1.79 | 79.95 ± 1.05 | 81.26 ± 1.24 | 82.80 ± 0.42 |
> | Grand-Clement et al. [1] | 76.16 ± 0.23 | 92.24 ± 1.97 | 76.53± 0.45      | 81.01 ± 1.74 | 75.53 ± 1.58 | 77.96 ± 1.29 | 76.20 ± 0.52 |
>
> [1] Grand-Clément, Julien, et al. "Interpretable machine learning for resource allocation with application to ventilator triage." arXiv preprint arXiv:2110.10994 (2021).
>
> > **Question 3** There are some confusions in the writing.
>
> Thank you for pointing those confusions out. We have corrected these typos in the revised manuscript.

---

### Meta-Review · Area_Chair_J3m2 · 2023-12-13

**Metareview:**

This paper has been assessed by three knowledgeable reviewers. Two of them scored it as rejectable (marginally) and one, who produced the shortest and least technically insightful and the least informative review, gave it an outlying maximum acceptance score. The reviewers generally liked the motivation: the paper approaches the problem of healthcare resource allocation with a RL technique. They however point out that the work is novel mainly in the context of the particular application scenario, but its technical novelty is limited. Of specific technical concerns, dependency of the method on the accurate state transition model has been highlighted, as well as hard to understand marginal effect of the particular set of assumptions used to construct the simulator - Even though the presented empirical results look interesting, it is not clear how much and in what way may they depend on the design choices of the state transition model and/or the simulator.  Additional gap involves the limitation imposed by budget constraint which impacts the extent of patient interactions that can be captured by the model. The reviewers would like it explored thoroughly. There were questions whether Q learning is the right framework to use in this particular application, and the analysis of alternatives such as model-based RL would be of interest. In summary, the paper shows a well-motivated and promising approach with a substantial room for further discussion to help the audience appreciate and understand marginal effects of design decisions. We encourage the authors to consider the constructive feedback in an attempt to solidify their story for a subsequent submission.

**Justification For Why Not Higher Score:**

The paper in its current form leaves multiple technical questions unanswered.

**Justification For Why Not Lower Score:**

n/a

---

### Decision · Program_Chairs · 2024-01-16

Reject